# Speech Emotion Recognition Based on Modified ReliefF

**DOI:** 10.3390/s22218152

**Published:** 2022-10-25

**Authors:** Guo-Min Li, Na Liu, Jun-Ao Zhang

**Affiliations:** College of Communication and Information Engineering, Xi’an University of Science and Technology, Xi’an 710600, China

**Keywords:** emotion recognition, feature selection, modified ReliefF, maximum information coefficient

## Abstract

As the key of human–computer natural interaction, the research of emotion recognition is of great significance to the development of computer intelligence. In view of the issue that the current emotional feature dimension is too high, which affects the classification performance, this paper proposes a modified ReliefF feature selection algorithm to screen out feature subsets with smaller dimensions and better performance from high-dimensional features to further improve the efficiency and accuracy of emotion recognition. In the modified algorithm, the selection range of random samples is adjusted; the correlation between features is measured by the maximum information coefficient, and the distance measurement method between samples is established based on the correlation. The experimental results on the eNTERFACE’05 and SAVEE speech emotional datasets show that the features filtered based on the modified algorithm significantly reduce the data dimensions and effectively improve the accuracy of emotion recognition.

## 1. Introduction

Affective computing enables computers to better recognize and express emotions. As the main research direction in the field of affective computing, emotion recognition is widely used in many fields, such as intelligent medicine [1], remote education and human-computer interaction [2]. Speech, as a major expressing form of emotion, contains rich emotional information. Therefore, speech emotion recognition (SER) has been a research focus in affective computing. SER refers to the technology of extracting emotional features from speech signals through computer processing to judge the type of human emotion, including preprocessing, feature extraction, and emotion classification. The focus of SER is to select and extract suitable features, and the quality of features determines the final accuracy of emotion recognition.

The features commonly used in SER mainly include sound quality features, spectral features, prosodic features, and corresponding statistical characteristics, such as the maximum, average, range, variance, etc. [3]. Prosodic features [4] describe the variation of speech, mainly including pitch frequency, speech energy, duration, etc. Spectral features describe the association between vocal movement and vocal channel change, mainly including cepstrum features (such as Mel cepstrum coefficient MFCC [5,6]) and linear spectral features (such as linear prediction coefficient LPC [7]). Sound quality features [8] reflect the vibration properties of sound and describe the clarity and identification of speech, including bandwidth, formant frequency, etc. In [9], the prosodic parameters of four type of emotions, such as anger, sadness, happiness and boredom, in the emotional database were studied and analyzed. In [10,11], prosodic features, such as energy, formants, and pitch, were extracted for speech emotion recognition. In [12], the Fourier parameter features were proposed from emotional speech signal for SER. In [13,14], the emotion recognition rate improved by concatenating Mel frequency cepstral coefficients with other feature sets, including energy and formant, pitch, and bandwidth. In [15], Mel frequency magnitude coefficient was extracted from speech signals, and multiclass SVM was used as the classifier to classify the emotions.

Up to now, scholars have proposed many effective emotional features, but these features often have high dimensions, with a large amount of redundancy. When using high-dimensional features directly for emotion analysis, it will prolong the model training time and impact the recognition performance. Therefore, it is necessary to select features to improve the efficiency and effectiveness of the model.

Feature selection refers to screening out a set of subsets from an existing dataset. The subsets meet certain criteria, while retaining the classification ability of original features as much as possible, removing irrelevant features, reducing the data dimension, and improving the model efficiency [16]. The feature selection method is divided into Wrapper [17] and Filter [18]. The wrapper algorithm directly uses the classification performance as the evaluation criterion of the feature importance. The subset selected by the strategy will eventually be used to construct the classification model. The filter method mainly uses distance, dependence, and other measurement criteria to calculate the implicit information in the features and gives the corresponding weight value of the features according to the calculation results. According to the weight, the important features under the criterion can be selected. This method directly obtains the implicit information of the features by mathematical calculation without involving the classifier; the strategy has high computational efficiency and can quickly eliminate non-critical features and remove noise features in the data.

In the process of speech emotion recognition, principal component analysis (PCA) was used for filtering feature selection to eliminate irrelevant features and improve the accuracy of classification in [19,20]. The feature selection method based on maximal relevance and minimal redundancy (MRMR) was used to evaluate the emotional features, which ensures the accuracy of emotional classification and effectively optimizes the feature set in [21]. The feature selection method based on CFS was used to evaluate the features and select the feature subset with high correlation with the category in [22]; it performs well on multiple emotional datasets. In [23,24], ReliefF algorithm was used to screen the emotional features of speech, which effectively reduced the feature dimension while ensuring the recognition rate.

The above feature selection methods have their own advantages. In contrast, the ReliefF algorithm has the characteristics of high efficiency and high precision. It can assign corresponding weights to the features according to the discrimination of features for limited samples in different categories. Therefore, many scholars have carried out related research using ReliefF algorithm combined with specific problems. For example, when studying the feature selection problem of hand gesture recognition, in [25], Minkowski distance was used to replace Euclidean distance to improve the selection method of nearest neighbor samples in ReliefF algorithm. In [26], the maximal information coefficient was used to replace the Euclidean distance to select the nearest neighbor samples, and the improved ReliefF algorithm was combined with the wrapper algorithm to automatically find the optimal feature subset. In [27], the features were sorted according to the classification performance of each feature, and the features with better performance were selected by setting the threshold. Then, the ReliefF algorithm was used to perform secondary screening on the features to achieve the purpose of dimensionality reduction.

In conclusion, the existing research combines the ReliefF algorithm with other methods, thus expanding its application scope and solving the feature selection problem of specific scenes. However, the ReliefF algorithm itself has defects, such as instability caused by randomness of the samples selected and redundancy among attributes. Therefore, in this paper, a modified ReliefF algorithm is proposed and applied to speech emotion recognition. The purpose is to select the optimal feature subset from the high-dimensional speech emotional features, reduce the feature dimensions, and improve the efficiency and accuracy of emotion recognition. The modified algorithm updates the selection range of random samples and the distance measurement between attributes. The block diagram of the speech emotion recognition system is shown in Figure 1.

## 2. Feature Extraction

### 2.1. Preprocessing

After the speech signal is digitized, it needs to be preprocessed to improve the quality of the speech. Speech is a non-stationary signal, but it can be regarded as a stationary signal in a small time period [28]. In order to obtain a short-term stable speech signal, it needs to be divided into frames, and there is a part of overlap between adjacent frames, which is called frame shift. Multiply the speech signal *s(n)* by a window function *w(n)* to obtain the framed speech:(1)$$s_{w}(n)=s(n)\times w(n),$$

### 2.2. Short Energy

Short energy, also called frame energy, is closely related to human emotional state. When people are emotionally excited, speech contains more energy; when people are depressed, speech contains less energy. Suppose the *i*th frame speech signal is *x_i_*(*m*), the frame length is *N*, and its short energy is: (2)$$E_{i}=\sum_{m=0}^{N-1}x_{i}^{2}(m),$$

### 2.3. Pitch Frequency

Pitch frequency is an influential feature parameter in SER, which represents the fundamental frequency of vocal cord vibration during vocalization. When a person is in a calm state, the pitch is relatively stable. When a person is in a happy or angry state, the pitch frequency becomes higher, and when a person is in a low mood, correspondingly the pitch frequency becomes lower. Usually, the autocorrelation function method is used to estimate pitch frequency. Suppose the *i*th frame of speech is *x_i_*(*m*), the frame length is *N*, and its short-time autocorrelation function is:(3)$$R_{i}(k)=\sum_{m=1}^{N-k}x_{i}(m)x_{i}(m+k),$$
where *k* represents the time delay. If the speech signal is periodic, its autocorrelation function is also periodic, and the period is the same as the speech period. On the integer multiple of the period, the autocorrelation function has a maximum value, the pitch period is estimated accordingly, and the inverse of the pitch period is the pitch frequency.

### 2.4. Formant

Formant reflects the physical characteristics of the vocal tract during vocalization. Different emotional speech cause different changes in the vocal tract, and the position of the formant frequency changes accordingly. The linear prediction method is usually used to estimate the formant parameters, and the transfer function of the vocal tract is expressed as:(4)$$H(z)=\frac{1}{A(z)}=\frac{1}{1-\sum_{k=1}^{p}a_{k}z^{-k}},$$
where *a_k_* represents the linear prediction coefficient, *p* represents the model order. Suppose *z_k_ = r_k_e^jθk^* is a root of *A(z)*, then the formant frequency is expressed as:(5)$$F_{k}=\frac{\theta_{k}}{2\pi T},$$

### 2.5. Fbank and MFCC

Fbank and MFCC are feature sets established by imitating the human auditory system. The human ear’s perception of frequency is not linear. In low frequency, the human ear’s perception of sound is proportional to the frequency of sound, but as the frequency increases, the ear’s perception of sound has a nonlinear relationship with frequency. On this basis, Mel frequency is introduced:(6)$$f_{Mel}=2595\times \mathrm{lg}\left(1+\frac{f}{700}\right),$$
where *f_Mel_* denotes the perception frequency in Mel, and *f* denotes the real frequency in Hz. Calculate discrete cosine transform on Fbank to obtain the Mel frequency cepstrum coefficient. Both MFCC and Fbank coefficients are commonly used feature parameters in the field of emotion recognition. The extraction process is shown in Figure 2.

## 3. Feature Selection

### 3.1. ReliefF

The ReliefF algorithm was proposed by Kononenko to solve the limitation that the Relief algorithm can only handle two-class problems [29]. The main idea is that the smaller the distance between samples of the same category and the greater the distance between samples of different categories, the more obvious the features’ effect on classification and the greater the weight. Conversely, the larger the distance between samples of the same category and the smaller the distance between samples of different categories, the weaker the feature’s effect on classification and the smaller the feature weight. The steps of the ReliefF algorithm are:(1)Initialize the weight vector *w* and the number of sampling times *m*;(2)Select a sample *R* randomly, and find *k* similar neighbors and heterogeneous neighbors, respectively. The distance between *R* and each neighbor *X_i_* on feature *f_r_* is calculated in (7):
(7)$$diff(f_{r},R,X_{i})=\frac{\left|R(f_{r})-X_{i}(f_{r})\right|}{\max(f_{r})-\min(f_{r})},$$(3)Update the weight of feature *f_r_*:
(8)$$w_{f_{r}}=-\sum_{j=1}^{k}diff(f_{r},R,H_{j})/(m\cdot k)+\sum_{C\neq class(R)}\left[\frac{P(C)}{1-P(class(R))}\sum_{j=1}^{k}diff(f_{r},R,M_{j}(C))\right]/(m\cdot k)$$
where *diff (f_i_*, *R*, *H_j_)* represents the distance difference between *R* and the *j*th neighbor of the same category *H_j_(j = 1*,*2*,…,*k)* on feature *f_r_*, *diff (f_i_*, *R*, *M_j_(C))* represents the distance difference between *R* and the *j*th neighbor of a different category *M_j_(C)(j =* 1,2,…,*k)* on feature *f_r_*, *P*(*C*) is the proportion of the samples of category *C* to total samples, and *P(class(R))* is the proportion of the category to which the sample *R* belongs.(4)Repeat the above steps *m* times, and the weight is averaged to obtain the final weight vector *w*.

### 3.2. Modified ReliefF

To ensure the stability of the feature selection algorithm, the samples are selected from each category on average, and the sampling range is the former *G* samples with the closest Euclidean distance to the center of the corresponding category. In addition, when the weight is updated, the maximal information coefficient (MIC) is used to measure the correlation between features, and the distance measurement method between sample features is established based on it.

MIC is a statistical method used to measure the dependence degree between variables [30]. Its essence is normalized mutual information, which has higher accuracy and universality. The mutual information of variables *x* and *y* is expressed as:(9)$$I(x;y)=\sum_{x}\sum_{y}p(x,y)\log_{2}\left(\frac{p(x,y)}{p(x)p(y)}\right),$$
where, *p*(*x,y*) represents joint probability of *x* and *y*, and *p*(*x*) and *p*(*y*) represent their probability density, respectively. Then, the maximum information coefficient is:(10)$$MIC(x;y)=\max\left\{\frac{I(x;y)}{\log_{2}\min(a,b)}\right\},$$
where *a* and *b* represent the number of grids divided on the *x* and *y* axes of the scatter diagram composed of vectors *x* and *y*, and *a* × *b < M^0.6^* (*M* is the number of samples). The size of the MIC value reflects the degree of correlation between features. The maximum information coefficient MIC (*f_r_*, *f_n_*) between the *r*th dimension feature *f_r_* and the *n*th dimension feature *f_n_* is marked as *s_rn_*; then, the correlation coefficient matrix between features is expressed as:(11)$$s=\left[\begin{array}{cccc} s_{11} & s_{12} & \cdots & s_{1N}\\ s_{21} & s_{22} & \cdots & s_{2N}\\ \vdots & \vdots & \cdots & \vdots \\ s_{N1} & s_{N2} & \cdots & s_{NN} \end{array}\right],$$
where *N* represents the total feature dimension. Define the distance measure between sample *X_i_* and *X_j_* over features *f_r_*:(12)$$dist(f_{r},X_{i},X_{j})=diff(f_{r},X_{i},X_{j})+1-\frac{1}{N-1}\sum_{n\neq r}^{N-1}s_{rn}\cdot diff(f_{n},X_{i},X_{j}),$$

Then, the distance between *X_i_* and *X_j_* is:(13)$$dist(X_{i},X_{j})=\frac{1}{N}\sum_{r=1}^{N}dist(f_{r},X_{i},X_{j}),$$

The specific process of the modified ReliefF algorithm is as follows:(1)Calculate the sample center of category *l*, and sort all samples in this category according to their distance to category center;(2)Randomly select sample *R* from the former *G* samples closest to the center of the category and repeat *m* times;(3)For current sample, find *k* neighbor samples of the same category and neighbor samples of different categories and calculate the distance between samples;(4)The weight is updated according to the ratio of the distance between the heterogeneous neighbors and the similar neighbors to assign a larger weight to the feature with large heterogeneous distance and small homogeneous distance, and vice versa, assign a smaller weight:
(14)$$w_{f_{r}}=\frac{D_{f_{r}}^{M}}{D_{f_{r}}^{H}},$$
where $D_{f_{r}}^{M}$ denotes the mean distance between sample *R* and the heterogeneous neighbors on feature *f_r_*, and $D_{f_{r}}^{H}$ denotes the mean distance between sample *R* and the similar neighbors on feature *f_r_*:(15)$$D_{f_{r}}^{H}=\sum_{j=1}^{k}dist(f_{r},R,H_{j})/(m\cdot k),$$
(16)$$D_{f_{r}}^{M}=\sum_{C\neq class(R)}\left[\frac{P(C)}{1-P(class(R))}\sum_{j=1}^{k}dist(f_{r},R,M_{j}(C))\right]/(m\cdot k),$$

(5)Repeat the above process for *L* categories and calculate the mean of the feature weights:


(17)
$${\bar{w}}_{f_{r}}=\sum_{l=1}^{L}w_{f_{r}}/L,$$


After the feature weights are obtained, the features are sorted in descending order according to the weights to obtain a feature set *F_IR_*.

The modified ReliefF algorithm considers the discrimination of different features to categories and the correlation between features in limited samples. In addition, the classification performance of each feature can be directly considered as weight to sort the features, and the obtained feature set *F_R_* is called performance-related features here. The features with better performance can be screened by setting a threshold. The two features are fused, and the fusion features are selected in combination with the model classification results to obtain a feature vector that can fully express the emotional state. The fusion features is expressed as in (18):(18)$$F_{F}=W_{R}*F_{R}+W_{IR}*F_{IR},$$
where *W_R_* represents the proportion of the features reordered by classification performance in the fusion features, *W_IR_* represents the proportion of the features reordered based on modified ReliefF weight.

## 4. Experiment and Results Analysis

The proposed method is validated on the eNTERFACE’05 dataset [31] and SAVEE dataset [32]. The eNTERFACE’05 dataset was performed by 42 subjects, with a total of 1287 audio files. The audio sampling frequency was 48 kHz, and the average duration was about 3 s. It includes six fundamental emotions: anger, disgust, fear, happiness, sadness and surprise. The SAVEE dataset was obtained by recording 120 emotional speeches by four subjects, with a sampling rate of 44.1 kHz. It includes seven types of emotions: anger, fear, joy, sadness, disgust, surprise, and neutrality, with a total of 480 speech files. From the dataset, 80% of each emotion was selected as the training data and the remaining 20% as the testing data.

Emotional features, including energy, first formant, pitch frequency, 13-order MFCC, delta and delta-delta MFCC, Fbank coefficients and their statistical features, including maximum, mean, variance, skewness, kurtosis, etc. were extracted, with a total of 235 dimensions. Support vector machine (SVM) and random forest (RF) classifier were used for emotion recognition.

Take the features of the eNTERFACE’05 dataset as an example to illustrate the necessity of feature selection. The emotion recognition accuracy of original features, i.e., unsorted features, under different dimensions is shown in Figure 3. Here, for the convenience of observation in the figure, the feature dimension is valued at an interval of 10; that is, the value range is 1:10:235.

It can be seen that with the increase of dimension, the amount of information contained in the feature set increases too, so the recognition accuracy curve generally shows an upward trend, but the curve is not monotonically increasing. For example, when the feature dimension increases from 70 to 90, the recognition rate correspondingly increases from 52% to 56%, while the features with dimensions between 90 and 100 lead to a 2% reduction in recognition rate, and the recognition rate is basically stable when the dimensions are 160 and 190. This suggests that not all features are beneficial to classification, and there may be adverse features or irrelevant features. When screening features, favorable features should be retained as much as possible, and unfavorable features and irrelevant features should be eliminated.

The feature weights are calculated by the feature selection algorithm, and the features are re-sorted according to the weights. The re-sorted features constitute different feature subsets according to different dimensions. The recognition rates of different feature subsets are compared and analyzed, including: (1) original features; (2) re-sorted features with PCA method; (3) re-sorted features with ReliefF method; (4) re-sorted features with MIC method; (5) re-sorted features with MRMR method; (6) re-sorted features with CFS method; and (7) fusion features based on the modified ReliefF method (Proposed). Among them, ReliefF algorithm is repeated 60 times, the number of nearest neighbors is 30, and the modified ReliefF algorithm is repeated 10 times in each category. The number of nearest neighbors is 30, the weight of performance-related features in the fusion features is 2, and the weight of modified ReliefF re-sorted features is 8.

The correlation curve between the recognition accuracy and the dimensions of feature subsets was analyzed, as shown in Figure 4 and Figure 5. It can be seen from the figure that the performance of features vary in different datasets and different classifiers, but in general, as the feature dimensions increase, the accuracy of emotion recognition increases accordingly. Different from the recognition rate curve of original features, when the emotion recognition rate based on feature selection increases to a certain extent, it will slowly decline or fluctuate within a certain range. The feature dimensions with the highest accuracy rate are the dimensions of the optimal feature subset. Among them, the fusion features based on the modified ReliefF algorithm has better performance, which has higher recognition accuracy and lower feature dimensions.

Table 1 shows the highest classification accuracy of the selected feature subsets of various methods, and Table 2 shows the minimum feature dimensions required for each method to achieve the final recognition accuracy. It can be seen from the table that the filtered features have better recognition performance, and the fusion features based on the modified algorithm performs best among all features. The average recognition rate of both datasets and the feature dimensions required improve at varying degrees compared with the original features. Especially for the eNTERFACE’05 dataset, when the feature dimensions is only 8.47% of the total dimensions, the fusion features reach the final recognition accuracy through SVM and RF. For the SAVEE dataset, the fusion features achieve the final recognition accuracy when the feature dimensions is 40 with SVM, and 70 with RF, accounting for 16.95% and 29.66% of the total dimensions, respectively.

The recognition accuracy of the fusion features based on modified ReliefF for each type of emotion was analyzed and compared with the recognition accuracy of the original features. The results are shown in Figure 6 and Figure 7. In general, the fusion features can distinguish each type of emotional state well, and in most cases, the fusion features perform better than the original features. It can be seen from Figure 6 that for the eNTERFACE’05 dataset, the recognition accuracy of the fusion features for “angry” and “surprise” achieves more than 90% through SVM, which is 4.65% and 6.97% higher than the original features, respectively. The best accuracy of the “surprise” state reaches 100% through RF. Moreover, the modified features greatly improved the recognition performance of the “disgust” state, and its accuracy is 18.61% higher than that of the original features.

From Figure 7, for the SAVEE dataset, the recognition accuracy of the fusion features for each type of emotion is better than the original features through SVM. Among them, the recognition accuracy of “disgust”, “fear”, “surprise”, and “neutral” reaches more than 90%, while the recognition accuracy of the original features for these emotional categories is only about 80%. With RF classifier, the fusion features effectively improve the recognition performance of “sadness” and “surprise”, and the recognition accuracy of “sadness” and “surprise” is 16.70% and 16.66% higher than the original features.

## 5. Conclusions

The quality of emotional features determines the accuracy of emotion recognition. The focus of this paper is to screen out the key features that are most discriminative for emotions from high-dimensional features and remove irrelevant features, reducing the model burden and improving recognition efficiency.

This paper put forward a modified feature selection algorithm to choose optimal speech emotion features. SVM and RF classifiers are applied to experimental analysis on eNTERFACE’05 and SAVEE datasets. The results show that the fusion features based on the modified algorithm can effectively solve the problem of high feature dimension in speech emotion recognition, and in the case of less feature dimension, better emotion classification results are obtained. On the eNTERFACE’05 dataset, the final recognition rate of the original features can be achieved by selecting at least 20 features from 236 features, which is 91.52% lower than the original feature dimensions, and the best classification accuracy of the modified method through SVM, 80.54%, was 4.67% higher than the original features, while the best classification accuracy through RF, 82.87%, was 6.61% higher than the original features. On the SAVEE dataset, the final recognition rate of the original features can be achieved by selecting at least 40 features from 236 features, which is 83.05% lower than the original feature dimensions, and the best classification accuracy of the modified method through SVM, 81.25%, was 9.38% higher than the original features, while the best classification accuracy through RF, 80.21%, was 3.13% higher than the original features.

At present, this paper mainly classifies emotions based on traditional emotional features. The next step is to study how to effectively integrate traditional features with deep features to further improve the effect of emotion recognition. In addition, this method can also be applied to feature selection problems in various fields such as pattern recognition.

## Figures and Tables

**Figure 1 sensors-22-08152-f001:**
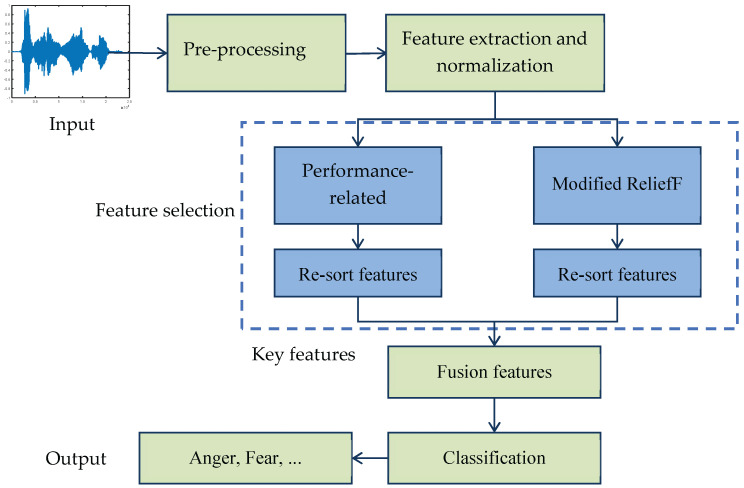
Block diagram of the speech emotion recognition.

**Figure 2 sensors-22-08152-f002:**
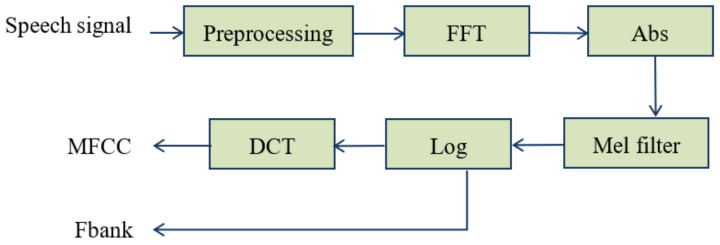
Extraction process of MFCC and Fbank.

**Figure 3 sensors-22-08152-f003:**
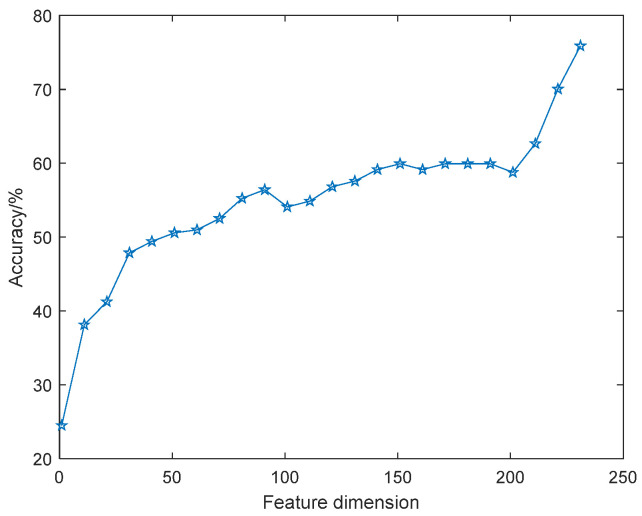
Classification accuracy of original features with feature dimension.

**Figure 4 sensors-22-08152-f004:**
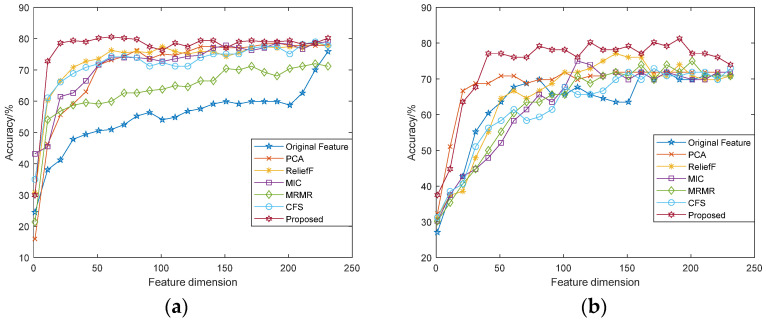
Accuracy comparison of feature subsets under different dimensions (SVM): (**a**) eNTERFACE’05 dataset; (**b**) SAVEE dataset.

**Figure 5 sensors-22-08152-f005:**
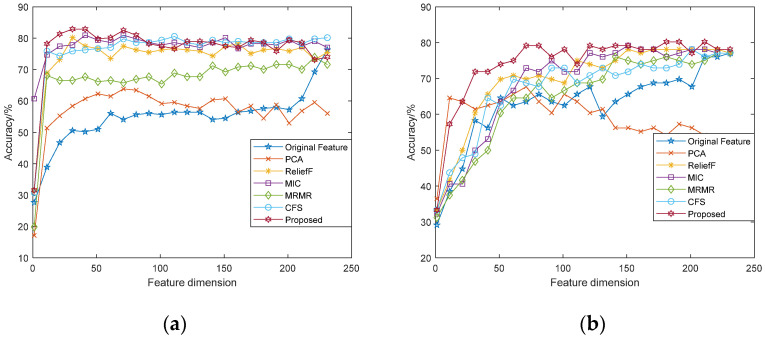
Accuracy comparison of feature subsets under different dimensions (RF): (**a**) eNTERFACE’05 dataset; (**b**) SAVEE dataset.

**Figure 6 sensors-22-08152-f006:**
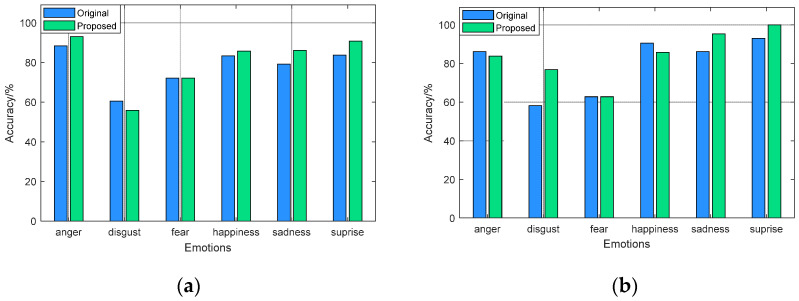
Recognition results on eNTERFACE’05: (**a**) SVM classifier; (**b**) RF classifier.

**Figure 7 sensors-22-08152-f007:**
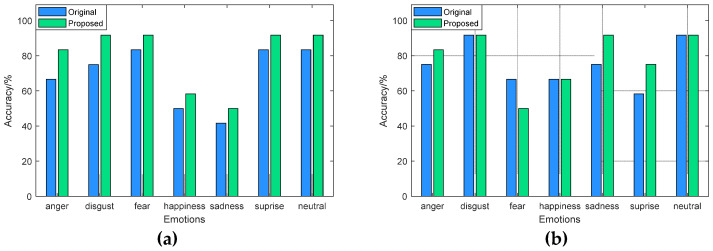
Recognition results on SAVEE: (**a**) SVM classifier; (**b**) RF classifier.

**Table 1 sensors-22-08152-t001:** Highest recognition rate for each method (%).

Dataset	Features	SVM	RF
eNTERFACE’05	Original Features	75.87	76.26
	PCA	78.59	63.81
	ReliefF	78.59	80.15
	MIC	78.98	80.93
	MRMR	71.98	73.93
	CFS	78.98	80.54
	Proposed	80.54	82.87
SAVEE	Original Features	71.87	77.08
	PCA	71.87	67.70
	ReliefF	77.08	78.12
	MIC	75.00	79.16
	MRMR	75.00	77.08
	CFS	72.91	78.12
	Proposed	81.25	80.21

**Table 2 sensors-22-08152-t002:** Minimum feature dimensions required for each method (%).

Dataset	Features	SVM	RF
eNTERFACE’05	Original Features	236	236
	PCA	80	236
	ReliefF	60	30
	MIC	140	20
	MRMR	236	236
	CFS	170	40
	Proposed	20	20
SAVEE	Original Features	236	236
	PCA	100	236
	ReliefF	100	150
	MIC	110	120
	MRMR	140	220
	CFS	150	200
	Proposed	40	70

## Data Availability

The datasets (eNTERFACE’05 and SAVEE datasets) used in this paper are available at http://www.enterface.net/results/ (accessed on 1 September 2022) and http://kahlan.eps.surrey.ac.uk/savee/Database.html (accessed on 1 September 2022).

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
