# Peer review of "Speech Emotion Recognition Based on Modified ReliefF"

_sensors, 2022, doi:10.3390/s22218152_

Round 1

Reviewer 1 Report

Your paper should have some numerical or experimental illustrations. In particular, the numerical experiments must have scientific value of their own. Furthermore, experimental comparisons with other approaches are strongly encouraged.

Your paper could improve in a number of areas such as a more thorough discussion of the design, development, testing and evaluation results; clarifying the key significance of the research contribution; ascertaining that the research fits the aims and scope of the journal; and a better command and flow of English writing throughout the paper.  

The references should be updated with the most recent in your paper's research field of relevance. I recommend the authors to consult the following survey and empirical papers to contextualize your findings. This should help the readers to understand the novelty of your work. 

Reliability of a distributed data storage system considering the external impacts,  IEEE Transactions on Reliability (2022), DOI: https://doi.org/10.1109/TR.2022.3161638/

Reliability of a distributed computing system with performance sharing, IEEE Transactions on Reliability (2022), DOI: https://doi.org/10.1109/TR.2021.3111031

Author Response

Thank you very much for your time involved in reviewing the manuscript and your very encouraging comments. We have revised this paper according to these comments and suggestions. A point-by-point response is provided, please see the attachment.

Reviewer 2 Report

In this paper, the authors proposed a modified ReliefF feature selection algorithm to screen out feature subsets with smaller dimensions with objective that better performance from high-dimensional features for emotion recognition.

The paper presents interesting results, which justify its publication after major corrections.

1- Authors should further discuss the results of section 4.

a) Present the minimum dimensions necessary for each method for an acceptable accuracy;

b) The results presented in figure 6 should be further discussed, especially for cases where the results are practically the same for both methods.

2- The conclusion must be improved, discussions about the percentage of accuracy increment in practical applications.

In this paper, the authors proposed a modified ReliefF feature selection algorithm to screen out feature subsets with smaller dimensions with objective that better performance from high-dimensional features for emotion recognition.

The paper presents interesting results, which justify its publication after major corrections.

1- Authors should further discuss the results of section 4.

a) Present the minimum dimensions necessary for each method for an acceptable accuracy;

b) The results presented in figure 6 should be further discussed, especially for cases where the results are practically the same for both methods.

2- The conclusion must be improved, discussions about the percentage of accuracy increment in practical applications.

Author Response

(The authors gave the same response as above.)

Reviewer 3 Report

The paper stuides speech emotion recognition based on modified relief. To improve this paper, I have some comments as follows:

(1)  For introduction section, Literature review should be more detailed and comprehensive. Authors should add more recent research progress to this part and give a brief introduction of the development history on your topic.

(2) It is better for authors to clarify your objectives of your study in the Introduction section.

(3) For your proposed method, advantages compared to other methods should be clarified.

(4) Reasons why you choose this method should be added to your manuscript.

(5) In the comparative study, authors can use other methods to analyze the same problem and highlight your method's accuracy.

(6) For conclusions, more detailed results should be presented.

(7) There are too many grammatical errors and typos. I recommend a professional editing service if possible. 

Author Response

(The authors gave the same response as above.)

Round 2

Reviewer 2 Report

The authors performed an extensive review of the paper, presenting a new version with the requested corrections. I consider that the current version can be accepted for publication.

Reviewer 3 Report

The paper can be accepted. All my questions have been addressed.